# Multisensory integration operates on correlated input from unimodal transient channels

## Cesare V Parise[1,2]*, Marc O Ernst[2,3]*

[1]Department of Psychology, University of Liverpool, Liverpool, United Kingdom; [2]Cognitive Neuroscience Department, University of Bielefeld (DE), Bielefeld, Germany; [3]Applied Cognitive Psychology, University of Ulm (DE), Ulm, Germany

## eLife assessment

This **important** study evaluates a model for multisensory correlation detection, focusing on the detection of correlated transients in visual and auditory stimuli. Overall, the experimental design is sound and the evidence is **compelling**. The synergy between the experimental and theoretical aspects of the article is strong, and the work will be of interest to both neuroscientists and psychologists working in the domain of sensory processing and perception.

***For correspondence:**
cesare.parise@liverpool.ac.uk (CVP);
marc.ernst@uni-ulm.de (MOE)

**Competing interest:** The authors declare that no competing interests exist.

**Abstract** Audiovisual information reaches the brain via both sustained and transient input channels, representing signals' intensity over time or changes thereof, respectively. To date, it is unclear to what extent transient and sustained input channels contribute to the combined percept obtained through multisensory integration. Based on the results of two novel psychophysical experiments, here we demonstrate the importance of the transient (instead of the sustained) channel for the integration of audiovisual signals. To account for the present results, we developed a biologically inspired, general-purpose model for multisensory integration, the multisensory correlation detectors, which combines correlated input from unimodal transient channels. Besides accounting for the results of our psychophysical experiments, this model could quantitatively replicate several recent findings in multisensory research, as tested against a large collection of published datasets. In particular, the model could simultaneously account for the perceived timing of audiovisual events, multisensory facilitation in detection tasks, causality judgments, and optimal integration. This study demonstrates that several phenomena in multisensory research that were previously considered unrelated, all stem from the integration of correlated input from unimodal transient channels.

## Introduction

Audiovisual stimuli naturally unfold over time, and their structures alternate intervals during which the signals remain relatively constant (such as the steady luminance of a light bulb) to sudden moments of change (when the bulb lights up). To efficiently process the temporal structure of incoming signals, the sensory systems of mammals (and other animal classes) rely on separate channels, encoding stimulus intensity through either sustained or transient responses (*Benucci et al., 2007*; *Kim et al., 2011*; *Breitmeyer and Ganz, 1976*; *Qin et al., 2007*; *Recanzone, 2000*; *Ikeda and Wright, 1972*). These channels are known to originate early in the processing hierarchy (*Recanzone, 2000*; *Ikeda and Wright, 1972*), with sustained ones responding with constant neural firing to static input intensity, whereas transient channels respond with increased firing to any variations in input intensity (i.e., both increments and decrements, *Figure 1a*). On a functional level, the response of sustained and transient

**Figure 1.** Sustained vs. transient channels. (**a**) Responses of sustained and transient channels to onset and offset step stimuli, and periodic signals comprising sequences of onsets and offsets. Note that while the sustained channels closely follow the intensity profile of the input stimuli, transient channels only respond to changes in stimulus intensity, and such a response is always positive, irrespective of whether stimulus intensity increases or decreases. Therefore, when presented with periodic signals, while the sustained channels respond at the same frequency as the input stimulus (frequency following), transient channels respond at a frequency that is twice that of the input (frequency doubling). (**b**) Synchrony as measured from cross-correlation between pairs of step stimuli, as seen through sustained (top) and transient (bottom) channels (transient and sustained channels are simulated using *Equations 1; 10*, respectively). Note how synchrony (i.e., correlation) for sustained channels peaks at zero lag when the intensity of the input stimuli changes in the same direction, whereas it is minimal at zero lag when the steps have opposite polarities (negatively correlated stimuli). Conversely, being insensitive to the polarity of intensity changes, synchrony for transient channels always peaks at zero lag. (**c**) Synchrony (i.e., cross-correlation) of periodic onsets and offset stimuli as seen from sustained and transient channels. While synchrony peaks once (at zero phase shift) for sustained channels, it peaks twice for transient channels (at zero and pi radians phase shift), as a consequence of its frequency-doubling response characteristic. (**d**) Experimental apparatus: participants sat in front of a black cardboard panel with a circular opening, through which audiovisual stimuli were delivered by a white LED and a loudspeaker. (**e**) Predicted effects of experiments 1 and 2 depending on whether audiovisual integration relies on transient or sustained input channels. The presence of the effects of interest in both experiments or the lack thereof indicates an inconclusive result, not interpretable in the light of our hypotheses.

channels represents distinct dynamic stimulus information. Specifically, sustained responses represent the intensity of the stimulus, while transient responses represent changes in stimulus intensity. In terms of frequency response, sustained channels can be characterized as low-pass temporal filters (*Equation 10*), highlighting the low-frequency signal components. Transient channels, on the other hand, have a

higher spectral tuning and can be characterized as band-pass temporal filters (*Equation 1*), signaling events, or moments of stimulus change (*Stigliani et al., 2017*).

Changing information over time is also critical for multisensory perception (*Stein, 2012*): when two signals from different modalities are caused by the same underlying event, they usually covary over time (like firecrackers' pops and blazes). A growing body of literature has now investigated human sensitivity and adaptation to temporal lags across the senses (*Vroomen and Keetels, 2010*), and it is well established that both multisensory illusions (*Sekuler et al., 1997*; *Samad et al., 2018*; *van Wassenhove et al., 2007*) and Bayesian-optimal cue integration (e.g., *Parise et al., 2012*) critically depend on synchrony and temporal correlation across the senses. However, multisensory integration does not operate on the raw physical signals; these are systematically transformed during transduction and early neural processing. Therefore to understand multisensory integration, it is critical to figure out how unisensory signals are processed before feeding into the integration stage. Surprisingly, this fundamental question has received little attention in multisensory research.

Current evidence, however, suggests a prominence of transient channels in the percept resulting from multisensory integration. For example, *Andersen and Mamassian, 2008* found that task-irrelevant increments or decrements in sound intensity equally facilitated the detection of both increments and decrements in the lightness of a visual display. Critically, such an effect only occurred when changes in the two modalities occurred in approximate temporal synchrony. Based on the independence of polarity of this crossmodal facilitation, where intensity increments and decrements produced similar perceptual benefits, the authors concluded that audiovisual integration relies primarily on unsigned transient stimulus information. The role of transient channels in audiovisual perception is further supported by fMRI evidence: *Werner and Noppeney, 2011* found that audiovisual interactions in the human brain only occurred during stimulus transitions and demonstrated that transient onset and offset responses could be dissociated both anatomically and functionally (see also *Herdener et al., 2009*). While these studies demonstrate the dominance of transient over sustained temporal channels in, for example, detection tasks as studied by *Andersen and Mamassian, 2008*, or the pattern of neural responses during passive observation of audiovisual stimuli as in *Herdener et al., 2009*, to date, it is still unknown to what extent transient and sustained channels affect the perceived timing of audiovisual events – such as the subjective synchrony of visual and auditory signals, which is arguably the primary determinant of multisensory integration.

To understand the effect of transient versus sustained channels in multisensory perception, we must first focus on the difference between their unimodal responses. For that, we can consider stimuli consisting of steps in stimulus intensity (e.g., *Andersen and Mamassian, 2008*). A schematic representation is shown in *Figure 1a*: while onset and offset step stimuli trigger identical unsigned transient responses, sustained responses differ across conditions. That is, given that sustained channels represent the magnitude of the stimulus, responses to onset and offset stimuli are negatively correlated. In signal processing, Pearson correlation is commonly used to assess the synchrony of two related signals, with a higher correlation representing higher synchrony (and similarity, *Wei, 2006*). Therefore, we can hypothesize that if multisensory time perception relies on sustained input channels, positively correlated audiovisual stimuli (e.g., onset or offset stimuli in both modalities) should be perceived as more synchronous than negatively correlated stimuli (i.e., onset in one modality, offset in the other). Conversely, if audiovisual synchrony relies on unsigned transients, positively and negatively correlated stimuli should appear equally synchronous (*Figure 1b*).

This hypothesis will be tested in experiment 1, where systematic differences in perceived synchrony based on whether audiovisual signals are positively or negatively correlated would provide evidence for a dominant role of sustained input channels in audiovisual temporal processing. A lack of systematic differences driven by stimulus correlation, however, would not necessarily imply a transient nature of audiovisual temporal processing: this would require additional evidence from an inverse experiment, one in which the transient hypothesis predicts an effect that the sustained hypothesis does not (*Figure 1e*). For that, we can consider audiovisual stimuli consisting of periodic onsets and offsets (e.g., square-wave amplitude modulation, see *Figure 1c*). While sustained channels respond at the same rate as the input, transient responses have a rate double that of the input signals. This phenomenon, known as frequency doubling, is commonly considered a hallmark of the contribution of transient channels in sensory neuroscience (e.g., *Kim et al., 2011*).

Frequency doubling, therefore, is a handle to assess the contribution of transient and sustained channels in audiovisual perception. Consider audiovisual stimuli defined by square-wave amplitude modulations with a parametric manipulation of crossmodal phase shift (*Figure 1c*). If audiovisual temporal integration relies on the correlation between sustained input channels, we can predict perceived audiovisual synchrony (i.e., correlation, *Wei, 2006*) to peak just once: at zero phase shifts. Conversely, if multisensory temporal integration relies on transient channels, perceived audiovisual synchrony should peak twice: at 0 and 180° phase lag (*Figure 1c*). Such a frequency-doubling phenomenon can be easily assessed psychophysically by measuring reported simultaneity as a function of audiovisual lags. Therefore, in a second psychophysical study, we rely on frequency doubling in audiovisual synchrony perception to assess whether multisensory integration relies on transient or sustained input channels.

## Results

### Experiment 1: Step stimuli

To probe the effect of the correlation between unimodal step stimuli on the perceived timing of audiovisual events, eight participants (age range 22–35 years, four females) observed audiovisual signals consisting of lightness and acoustic intensity increments (on-step) and decrements (off-step). On-steps and off-steps were paired in all possible combinations, giving rise to four experimental conditions (both modalities on, both off, vision on with audio off, and vision off with audio on, see *Figure 1b*). The lag between visual and acoustic step events was parametrically manipulated using the method of constant stimuli (15 steps, between –0.4 and 0.4 s). After the stimulus presentation, participants performed a temporal order judgment (TOJ, which event came first, sound or light?) or a simultaneity judgment (SJ, were the stimuli synchronous or not?). TOJ and SJ tasks were run in different sessions occurring on different days. Although our original hypotheses (see, *Figure 1b–e*) do

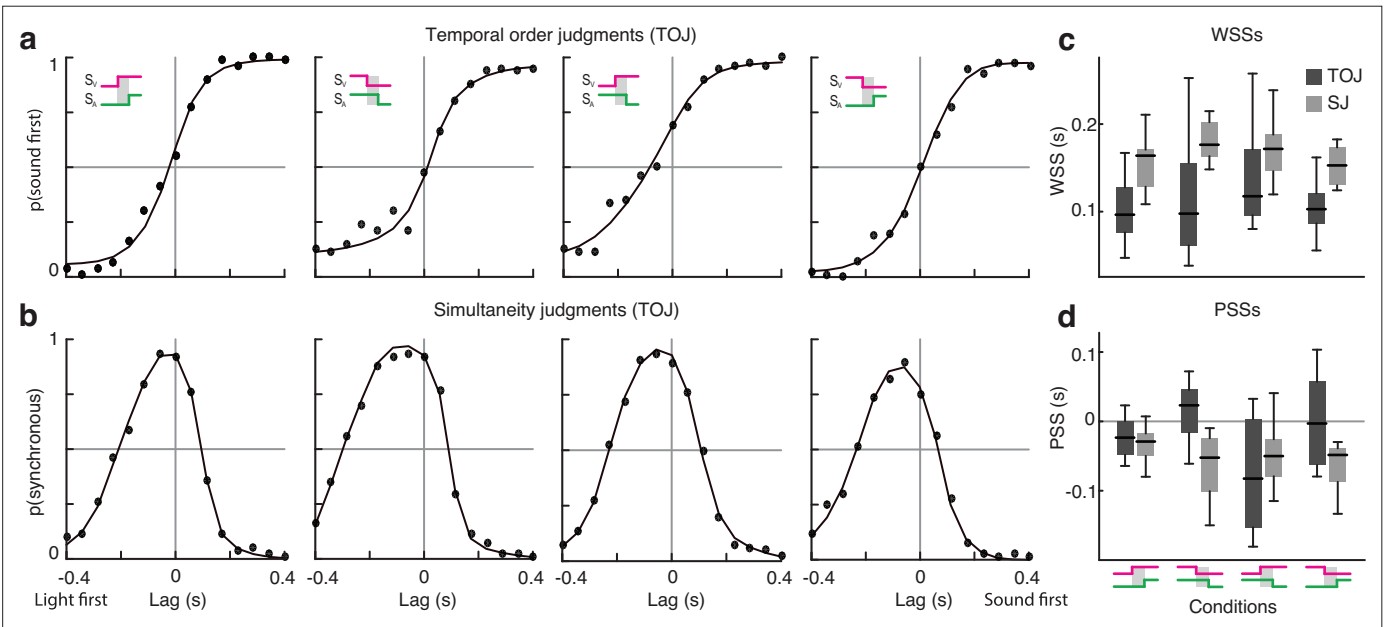

**Figure 2.** Experiment 1: results. (**a**) Responses in the temporal order judgment (TOJ) task and psychometric fits (averaged across participants) for the four experimental conditions. (**b**) Responses in the simultaneity judgment (SJ) task and psychometric fits (averaged across participants) for the four experimental conditions. Each dot in (**a**, **b**) corresponds to 80 trials. (**c**) Window of subjective simultaneity for each condition and task. (**d**) Point of subjective simultaneity for each condition and task.

The online version of this article includes the following source data and figure supplement(s) for figure 2:

**Source data 1.** Data of the temporal order judgment (TOJ) task.

**Source data 2.** Data of the simultaneity judgment (SJ) task.

**Figure supplement 1.** Results and psychometric fits of experiment 1.

not make specific predictions for the TOJ task, for completeness we run such an experiment anyway, as its inclusion provides a more stringent test for our model (see 'Modeling' section).

The audiovisual display used to deliver the stimuli consisted of a white LED in an on- (high-luminance) and off-state (low-luminance), and acoustic white noise, also in either on- (quiet) or off-state (loud). Sounds came from a speaker located behind the LED, and both the speaker and the LED were controlled using an audio interface to minimize system delay (*Figure 1d*, see *Parise and Ernst, 2016*). A white, sound transparent cloth was placed in front of the LED so that when the light was on, participants saw a white disk of 13° in diameter. Overall, each participant provided 600 responses in the TOJ task (15 lags, 10 repetitions, and 4 conditions) and additional 600 responses in the SJ task (again, 15 lags, 10 repetitions, and 4 conditions). The experiment was run in a dark, sound-attenuated booth, and the position of participants' heads was controlled using a chin- and a headrest. Participants were paid 8 euros/hr. The experimental procedure was approved by the ethics committee of the University of Bielefeld (ref. no. 2015-136) and was conducted in accordance with the Declaration of Helsinki.

## Results

To assess whether different combinations of on-step and off-step stimuli (and correlation thereof) elicit measurable psychophysical effects, we estimated both the point and the window of subjective simultaneity (PSS and WSS, i.e., the delay at which audiovisual stimuli appear simultaneous, and the width of the window of simultaneity). For that, we fitted psychometric curves to both TOJ and SJ data, independently for each condition. Specifically, following standard procedures (*Parise and Spence, 2009*), TOJs were statistically modeled as cumulative Gaussians, with four free parameters (intercept, slope, and two asymptotes, *Figure 2a*). The PSS was calculated as the lag at which TOJs were at chance level, whereas the WSS was calculated as the half-difference between the lags eliciting 0.75 and 0.25 probability of audio-first responses. The SJs data, instead, were modeled as the difference of two cumulative Gaussians (*Yarrow et al., 2011*), leading to asymmetric bell-shaped psychometric functions (*Figure 2b*). The PSS was calculated as the lag at which perceived simultaneity was maximal, whereas the WSS was calculated as the half-width at half-maximum. Results are summarized in *Figure 2c and d* (see *Figure 2—figure supplement 1* for individual data, and 'Materials and methods' for a correlation analysis of the PSS and WSS measured from TOJs vs. SJs).

To assess whether the four experimental conditions differ in the PSS and WSS, we run both nonparametric Friedman tests, with Bonferroni correction for multiple testing, and Bayesian repeated-measures ANOVA. Neither PSS nor WSS statistically differed across conditions, and this was true for both the TOJ and SJ data. Results are summarized in *Appendix 1—table 2*.

## Conclusion

The lack of a difference across conditions found in experiment 1 implies that the on-step and off-step stimuli induced similar perceptual responses. Based on our original hypothesis, the present results argue against the dominance of sustained input channels in the perception of audiovisual events, as the synchrony (i.e., Pearson correlation, *Wei, 2006*) of sustained signals would otherwise have been affected by the crossmodal combination of on-step and off-step stimuli. As previously mentioned, while a lack of difference across conditions in experiment 1 is necessary to infer a dominance of transient channels in audiovisual time perception, this evidence is not sufficient on its own. For that, we would need additional evidence in the form of the presence of systematic effects, which are predicted from the operating principles of the transient channels, but not for sustained ones. Experiment 2 was designed to fulfill such a requirement as it predicts a frequency-doubling effect in perceived synchrony for transient but not for sustained input channels.

## Experiment 2: Periodic stimuli

Participants observed a periodic audiovisual stimulus consisting of a square-waved intensity envelope and performed a force-choice SJ task. The carrier visual and auditory stimuli consisted of pink noise, delivered by a speaker and an LED (*Figure 1d*), which were switched on and off periodically in a square-wave fashion with a period of 2 s for a total duration of 6 s (so that three full audiovisual cycles were presented on each trial, *Figure 3a*). To prevent participants from just focusing on the start and endpoint of the signals, during the intertrial interval, the visual and auditory stimuli were set to a

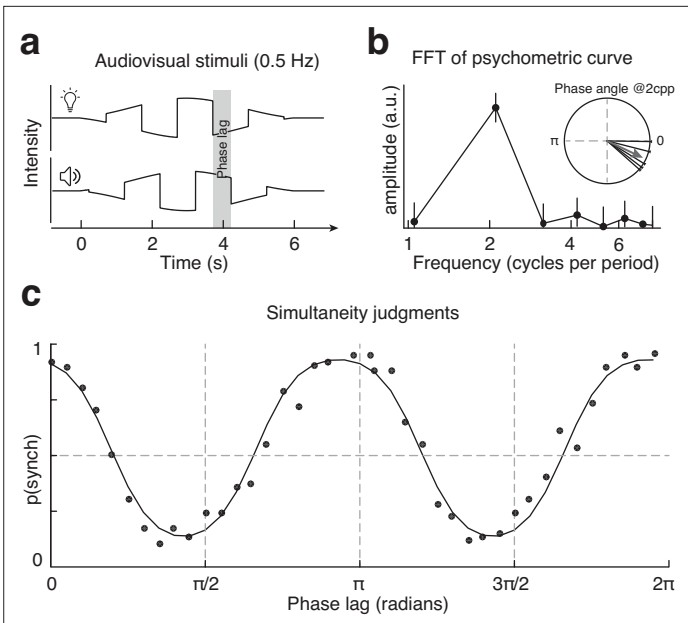

**Figure 3.** Experiment 2: stimuli and results. (**a**) Schematic representation of the periodic stimuli. (**b**) Frequency domain representation of the psychometric function: note the amplitude peak at two cycles per period. Errorbars represent the 99% CIs. The inset represents the phase angle of the two cycles per period frequency component for each participant (thin lines) and the average phase (arrow). (**c**) Results of experiment 2 and psychometric fit, averaged across all participants. Each dot corresponds to 75 trials.

The online version of this article includes the following source data and figure supplement(s) for figure 3:

**Source data 1.** Data of the simultaneity judgment (SJ) task.

**Figure supplement 1.** Results and psychometric fits of experiment 2.

pedestal intensity level, so that during the trial the square-wave modulation was gradually ramped on and off, following a raised cosine profile with a duration of 6 s (*Figure 3a*; see also *Figure 6—figure supplement 5C*).

The lag across the sensory signals consisted of relative phase shifts of the two square waves, while the raised cosine window remained constant (and synchronous across the senses). Audiovisual phase shift was manipulated according to the method of constant stimuli, and a full cycle was sampled in 40 steps. Each lag was presented 15 times to yield a total of 600 trials per participant. Besides the relative phase shifts across the multisensory signals, also the phase offset of the stimulus as a whole varied pseudorandomly across trials (spanning a full period sampled in 15 steps, one per repetition). Therefore, each time a given lag (phase shift) was tested, also the phase offset of the signals changed. Given that we expect the frequency-doubling effect to be strong in size (i.e., synchrony at pi-phase shift should approach 0 under the sustained hypothesis, and 1 under the transient hypothesis), and that we collected a large number of responses (n = 600 per observer), a pool of five participants (age range 25–35 years, three females) was large enough to reliably assess its presence or lack thereof. Participants were paid 8 euros/hr, and the experimental procedure was approved by the ethics committee of the University of Bielefeld and was conducted in accordance with the Declaration of Helsinki.

## Results

Due to the periodic nature of the stimuli and hence of the experimental manipulation of lag, the resulting psychometric functions are also expected to be periodic, with an alternation of phase shifts yielding higher and lower reported synchrony. In this context, the evidence of frequency doubling can be measured from the number of oscillations in perceived simultaneity for a full cycle of phase shifts between the audiovisual stimuli.

Given the periodicity of the stimuli, it is natural to analyze the psychometric functions in the frequency domain. Therefore, to get a nonparametric and assumption-free estimate of the frequency

of oscillations in the data, we ran a Fourier analysis on the empirical psychometric curves. If human responses are driven by transient input channels, we predict a peak at two cycles per period (cpp., i.e., frequency doubling), otherwise, there should be a single peak at 1 cpp. The power spectrum of the psychometric functions shows a sharp peak at a frequency of 2 cpp in all observers, thereby indicating the presence of the hypothesized frequency-doubling effect (*Figure 3b*, *Figure 3—figure supplement 1*). To assess whether the amplitude at 1 and 2 cpp is statistically different at the individual observer level, we used a bootstrap procedure to estimate the CIs of the response spectrum. For that, we used the binomial distribution and simulated 50,000 psychometric functions, on which we performed a frequency analysis to obtain the 99% CIs of the amplitudes. The results for both the aggregate observer and the individual data show a clear separation between the CIs of the amplitude at the frequencies of 1 and 2 cpp and demonstrate that the 2 cpp is indeed the dominant frequency, whereas the amplitude at 1 cpp is close to 0 and it is no different from the background noise (i.e., higher harmonics; see *Figure 3b*, *Figure 3—figure supplement 1*).

Given that the frequency analyses revealed a 2 cpp peak for all participants, we used this information to fit a sinusoidal psychometric function to the psychophysical data. Under the assumption of late Gaussian noise, the psychometric function can be written as

$$p\left(synch \mid \phi\right) = normcdf\left(\alpha + \beta \cdot sin\left(f \cdot \phi + \theta\right)\right),$$

where *normcdf* represents the cumulative normal distribution function, the parameter $\alpha$ is the bias term, $\beta$ the sensitivity, $\phi$ the phase-lag (range = [0, $2\pi$]), $f$ the frequency of oscillations, and $\theta$ is the phase offset (this shifts peak synchrony toward either positive or negative phases). Based on the Fourier analyses, we fixed the frequency ($f$) to 2 cpp, $\theta$ to –0.441 radians, and kept the linear coefficients – $\alpha$ and $\beta$ – as free parameters, which were fitted to the psychophysical data using an adaptive Bayesian algorithm (*Acerbi and Ma, 2017*).

Overall, the fitted psychometric curves match the empirical data with a high goodness of fit (median $r^2$ = 0.9321). Such an agreement between psychometric curves and empirical data achieved with the frequency parameter constrained by the Fourier analyses further indicates a reliable frequency-doubling effect in all our participants.

## Conclusion

The results of experiment 2 clearly demonstrate the existence of a frequency-doubling effect in the perceived simultaneity of periodic audiovisual stimuli. Given that the frequency-doubling effect is only expected to occur if audiovisual synchrony is computed over unsigned transient input channels, the present results support a dominance of transient input channels in multisensory time perception. These results complement the conclusions of experiment 1 and together demonstrate the key role of transient input channels in audiovisual integration (see *Figure 1E*).

## Modeling

To account for multisensory integration, we have previously proposed a computational model, the multisensory correlation detector (MCD, *Parise and Ernst, 2016*) that exploits the temporal correlation between the senses to solve the correspondence problem, detect simultaneity and lag across the senses, and perform Bayesian-optimal multisensory integration. Based on the Hassenstein–Reichardt motion detector (*Hassenstein and Reichardt, 1956*; *Fujisaki and Nishida, 2005*; *Fujisaki and Nishida, 2007*), the core of the MCD is composed of two mirror-symmetric subunits, each multiplying visual and auditory input after applying a low-pass filter to each of them. As a consequence of this asymmetric filtering, each subunit is selectively tuned to different temporal order of the signals (i.e., vision vs. audition lead). The outputs of the two subunits are then combined in different ways to detect the correlation and lag of multisensory signals, respectively. Specifically, correlation is calculated by multiplying the outputs of the subunits, hence producing an output ($MCD_{Corr}$) whose magnitude represents the correlation between the signals (*Figure 4c*). Temporal lag is instead detected by subtracting the outputs of the subunits, like in the classic Hassenstein–Reichardt detector (*Hassenstein and Reichardt, 1956*). This yields an output ($MCD_{Lag}$) with a sign that represents the temporal order of the signals (*Figure 4b*). While a single MCD unit can only perform temporal integration of multisensory input, a population of MCD units, each receiving input from spatially tuned receptive fields (*Figure 4d*), followed by divisive normalization (*Figure 4e*, see *Ohshiro et al., 2011*), can

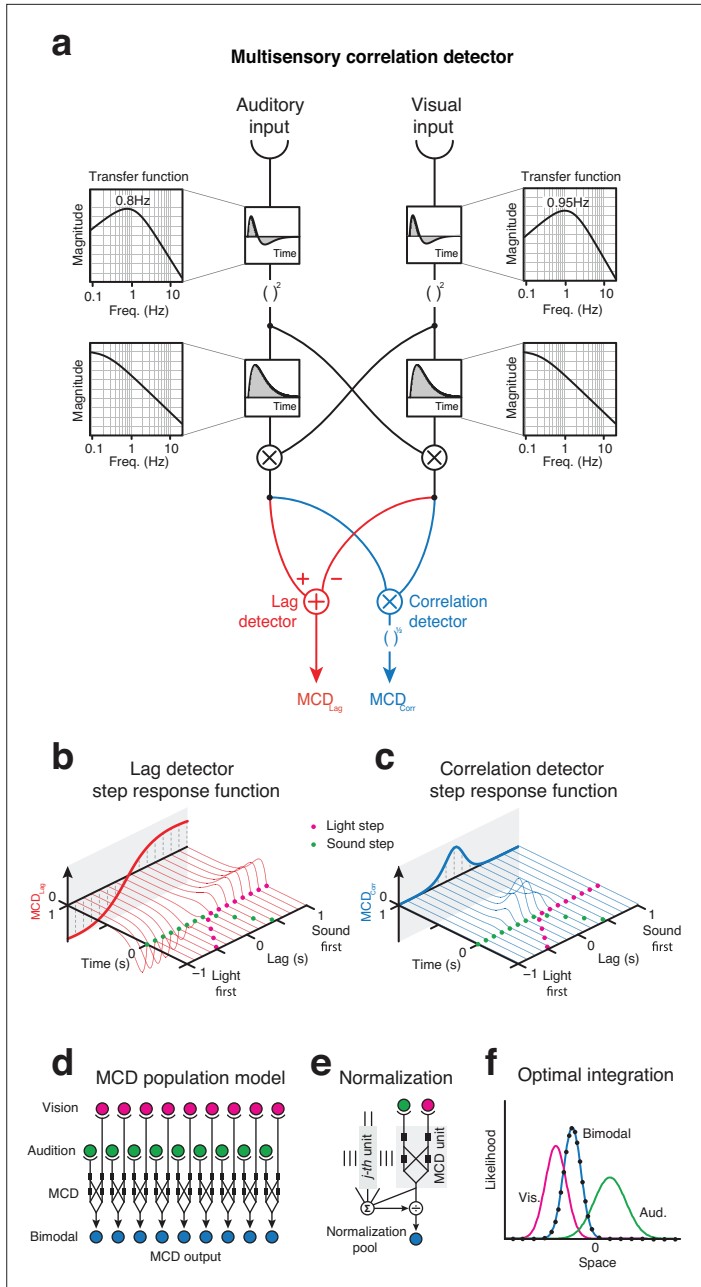

**Figure 4.** Multisensory correlation detector (MCD) model. (**a**) Model schematics: the impulse-response functions of the channels are represented in the small boxes, and call-outs represent the transfer functions. (**b**) Lag detector step responses as a function of the lag between visual and acoustic steps. (**c**) Correlation detector responses as a function of the lag between visual and acoustic steps. (**d**) Population of MCD units, each receiving input from spatiotopic receptive fields. (**e**) Normalization, where the output of each unit is divided by the sum of the activity of all units. (**f**) Optimal integration of audiovisual spatial cues, as achieved using a population of MCDs with divisive normalization. Lines represent the likelihood functions for the unimodal and bimodal stimuli; dots represent the response of the MCD model, which is indistinguishable from the bimodal likelihood function.

perform Bayesian-optimal spatial cue integration (e.g., see *Alais and Burr, 2004*) for audiovisual source localization (*Figure 4f*, see *Parise and Ernst, 2016* and *Parise, 2025*, for details).

In its original form, the input to the MCD consisted of sustained unimodal channels, modeled as low-pass temporal filters (see also, *Burr et al., 2009*; *Yarrow et al., 2011*). Although such a model could successfully account for the integration of trains of audiovisual impulses, the MCD cannot replicate

the present results: for that, we first need to feed the model with unsigned transient unimodal input channels. Therefore, we replaced the front-end low-pass filters with band-pass temporal filters (to detect transients) followed by a squaring non-linearity (to get the unsigned transient; *Stigliani et al., 2017*), and tested the model against the results of both experiments 1 and 2, plus a variety of previously published psychophysical studies.

The equations of the revised MCD model are reported in the 'Materials and methods'. Just like the original version, the revised MCD model has three free parameters, representing the time constants of the two band-pass filters (one per modality) and that of the low-pass temporal filters of the subunits of the detector. Given that the tuning of the time constants of the model depends both on physiological constraints and on the temporal profile of the input stimuli (see *Pesnot et al., 2022*), here we used the data from experiments 1 and 2 to constrain the temporal constants for the simulation of datasets consisting of either step stimuli or variations thereof (e.g., periodic stimuli), while for the simulation of experiments relying on stimuli with faster temporal rates, we constrained the temporal constants using data from *Parise and Ernst, 2016*. Details on parameter fitting are reported in the 'Materials and methods'. Given that in a previous study (*Parise and Ernst, 2016*) we have already shown that

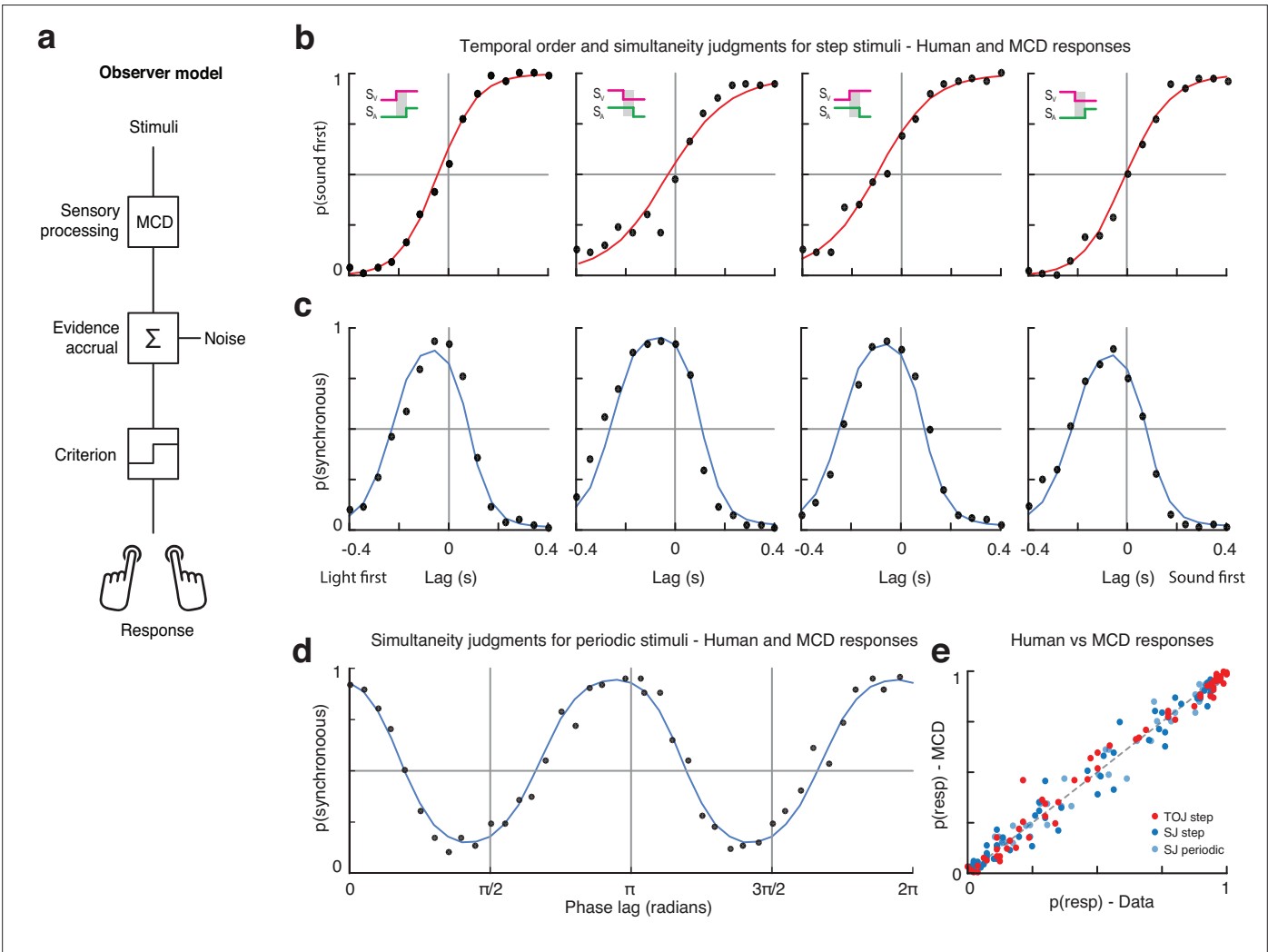

**Figure 5.** Multisensory correlation detector (MCD) simulations of experiments 1 and 2. (**a**) Schematics of the observer model, which receives input from one MCD unit to generate a behavioral response. The output of the MCD unit is integrated over a given temporal window (whose width depends on the duration of the stimuli) and corrupted by late additive noise before being compared to an internal criterion to generate a binary response. Such a perceptual decision-making process is modeled using a generalized linear model (GLM), depending on the task, the predictors of the GLM were either $\overline{MCD_{corr}}$ (*Equation 8*) or $\overline{MCD_{lag}}$ (*Equation 9*). (**b**) Responses for the temporal order judgment (TOJ) task of experiment 1 (dots) and model responses (red curves). (**c**) Responses for the simultaneity judgment (SJ) task of experiment 1 (dots) and model responses (blue curves). (**d**) Experiment 2 human (dots) and model responses (blue curve). (**e**) Scatterplot of human vs. model responses for both experiments.

alternative models for audiovisual integration are not flexible enough to reproduce the wide gamut of psychophysical data successfully accounted for by our model, here we only consider the MCD model (in its revised form). A MATLAB script with the full implementation of the revised MCD model is available in the 'Materials and methods'.

## Simulation of experiments 1 and 2

To test whether an MCD that receives input from unimodal transient channels could replicate the results of experiments 1 and 2, we fed the stimuli to the detector and used the $\overline{MCD_{corr}}$ (*Equation 8*) output to model SJs, and the $\overline{MCD_{lag}}$ (*Equation 9*) for the TOJs (see *Parise and Ernst, 2016*). Given that the psychophysical data consisted of response probabilities (i.e., probability of 'synchronous' responses for the SJ and probability of 'audio first' responses in the TOJ), whereas the outputs of the model are continuous variables expressed in arbitrary units, we used a GLM with a probit link function to transform the output of the MCD into probabilities (*Figure 5a*, see also *Parise and Ernst, 2016* for a detailed description of the approach). The linear coefficients (i.e., slope and intercept) were fitted separately for each condition and task so that each psychometric curve had two free parameters for the GLM. The three parameters defining the temporal constants of the MCD model, instead, were fitted using all data from experiments 1 and 2 combined using an adaptive Bayesian algorithm (*Acerbi and Ma, 2017*).

Overall, the model could tightly replicate the results of our experiments (*Figure 5b–d*): the Pearson correlation between the psychophysical data and model responses computed across all conditions and participants was 0.99 for the SJ task of experiment 1, 0.99 for the TOJ task of experiment 1, and 0.97 for experiment 2 (*Figure 5e*, see *Appendix 1—table 1*). For comparison, the correlation between the data and the psychometric fits for experiment 1 (i.e., cumulative Gaussians for the TOJs and difference of two cumulative Gaussians for the SJ) was 0.97 and 0.98 for experiment 2; however, the psychometric fits required nearly twice as many free parameters compared to the model fits and do not account for the generative process that give rise to the observed data. Importantly, just like human responses, the responses of the revised MCD model were nearly identical across all conditions in experiment 1, whereas in experiment 2 they displayed a clear frequency-doubling effect. Furthermore, unlike the psychometric fits, which require to specify a priori the shape of the fitting function (e.g., sigmoid, bell, or periodic), the MCD model provides an output without specifying a priori any shape for the psychometric function. As shown in *Figure 5b–d*, the model naturally captures the shape of the psychometric data. For instance, the same $\overline{MCD_{corr}}$ output could generate bell-shaped response distributions in experiment 1 and sinusoidal responses in experiment 2, purely based on the features of the input signals (such as its periodicity, or lack thereof). Therefore, taken together, the present simulations demonstrate that multisensory perception does indeed operate on correlated input from unimodal transient channels.

## Validation of the MCD through simulation of published results

Although the simulations of experiments 1 and 2 demonstrate that an MCD unit that receives input from transient unimodal channels is capable of reproducing the present psychophysical results, it is important to assess the generalizability of this approach. Therefore, we relied on the previous literature on multisensory perception of correlation, simultaneity, and lag, as well as on the performance on crossmodal detection tasks, to validate our computational framework and assess its generalizability (by comparing the MCD responses against human performance on tasks not originally designed around our model). To this end, we selected a series of studies that employed parametric manipulations of the temporal structure of the signals, simulated the stimuli, and used the MCD model to predict human performance. If an MCD unit that receives input from transient unimodal channels is indeed the elementary computational unit for multisensory temporal processing, we should be able to reproduce all of the earlier findings on multisensory perception with our revised MCD model. A summary of our simulations, listing the sample size, number of observers, and Pearson correlation of MCD and human responses, is reported in *Appendix 1—table 1*.

## Causality and temporal order judgments for random sequences of audio-visual impulses

When we first proposed the MCD model (*Parise and Ernst, 2016*), we tested it against a psychophysical experiment in which participants were presented with a random sequence of five clicks and five flashes over an interval of 1 s (*Figure 6—figure supplement 1a*), and had to report whether the signals appeared to share a common cause (causality judgments) and which modality came first (TOJ). To test whether the revised MCD model could also account for these previous findings, we fed the same stimuli to the model and used the $\overline{MCD_{corr}}$ (*Equation 8*) output to simulate causality judgments, and $\overline{MCD_{lag}}$ (*Equation 9*) to simulate TOJs. Given that the stimuli in this experiment had a much higher temporal rate than the stimuli used in experiments 1 and 2, the temporal constants of the MCD were set as free parameters that we fitted to the experimental data. Due to the stochastic nature of the stimuli, we analyzed the data using reverse correlation (*Figure 6—figure supplement 1b*). The temporal constants of the MCD were fitted to maximize the Pearson correlation between the empirical and simulated responses, for both causality and TOJs (for details on modeling and reverse correlation analyses, see *Figure 6—figure supplement 1b* and *Parise and Ernst, 2016*).

Overall, the model faithfully reproduced the experimental data, and the Pearson correlation between empirical and simulated classification images was 0.99 (*Figure 6a and b*). This result closely replicates the original simulations (*Parise and Ernst, 2016*) performed with a version of the MCD that instead received input from sustained unimodal temporal channels (not transient as the current model). Given the differences between transient and sustained temporal channels, it may seem surprising that the two models are in such close agreement in the current simulation. However, it is important to remember that the stimuli in these experiments consisted of impulses (clicks and flashes), and the impulse response function of the sustained and transient channels as modeled in this study is indeed very similar. Therefore, such an agreement between the responses of the original and the revised versions of the MCD is expected. Besides proving that the revised model can replicate previous results, this simulation allowed us to estimate the temporal constants of the MCD for trains of clicks and flashes, which was necessary for the next simulations.

## Causality judgments for random sequences of audiovisual impulses with high temporal rates

To identify the temporal features of the stimuli that promote audiovisual integration, *Locke and Landy, 2017*, experiment (2) ran a psychophysical experiment that was very similar to the causality judgment task described in the previous section. Although the stimuli in the two studies were nearly identical, the temporal sequences used by *Locke and Landy, 2017* had a considerably higher temporal rate (range 8–14 impulses/s), longer duration (2 s), and a more controlled temporal structure (*Figure 6—figure supplement 2*). Participants observed the stimuli and performed a causality judgment task, whose results demonstrated that the perception of a common cause depended both on the correlation in the temporal structure of auditory and visual sequences (*Figure 6C*, left) and the maximum lag between individual clicks and flashes (*Figure 6C*, right).

Given the similarity of this study with the causality judgment of *Parise and Ernst, 2016*, we followed the same logic described above to perform reverse correlation analyses (without smoothing the cross-correlograms). Moreover, the MCD simulations were performed with the same temporal constants used for the simulations of *Parise and Ernst, 2016*, so that this experiment is simulated with a fully constrained model, with zero free parameters. Unlike our previous experiment, however, the stimuli used by *Locke and Landy, 2017* varied in temporal rate, and hence also the number of clicks and flashes differed across trials. Given that the MCD is sensitive to the total stimulus energy, we normalized the model responses by dividing the $\overline{MCD_{corr}}$ (*Equation 8*) output by the rate of the stimuli.

Reverse correlation analyses were performed at the single subject level using both participants and model responses (see *Parise and Ernst, 2016* for details on how the continuous model output of the MCD was discretized into a dichotomous variable), and the average data is shown in *Figure 6c* (left panel). Overall, the MCD model could near-perfectly predict the empirical classification image (Pearson correlation >0.99). Besides the reverse correlation analyses, *Locke and Landy, 2017* measured how the maximum lag between individual clicks and flashes affected the perceived common cause of audiovisual sequences. The results, shown in *Figure 6c* (right panel), demonstrate that the perception of a common cause decreased with increasing maximum audiovisual lag. Once again, the MCD

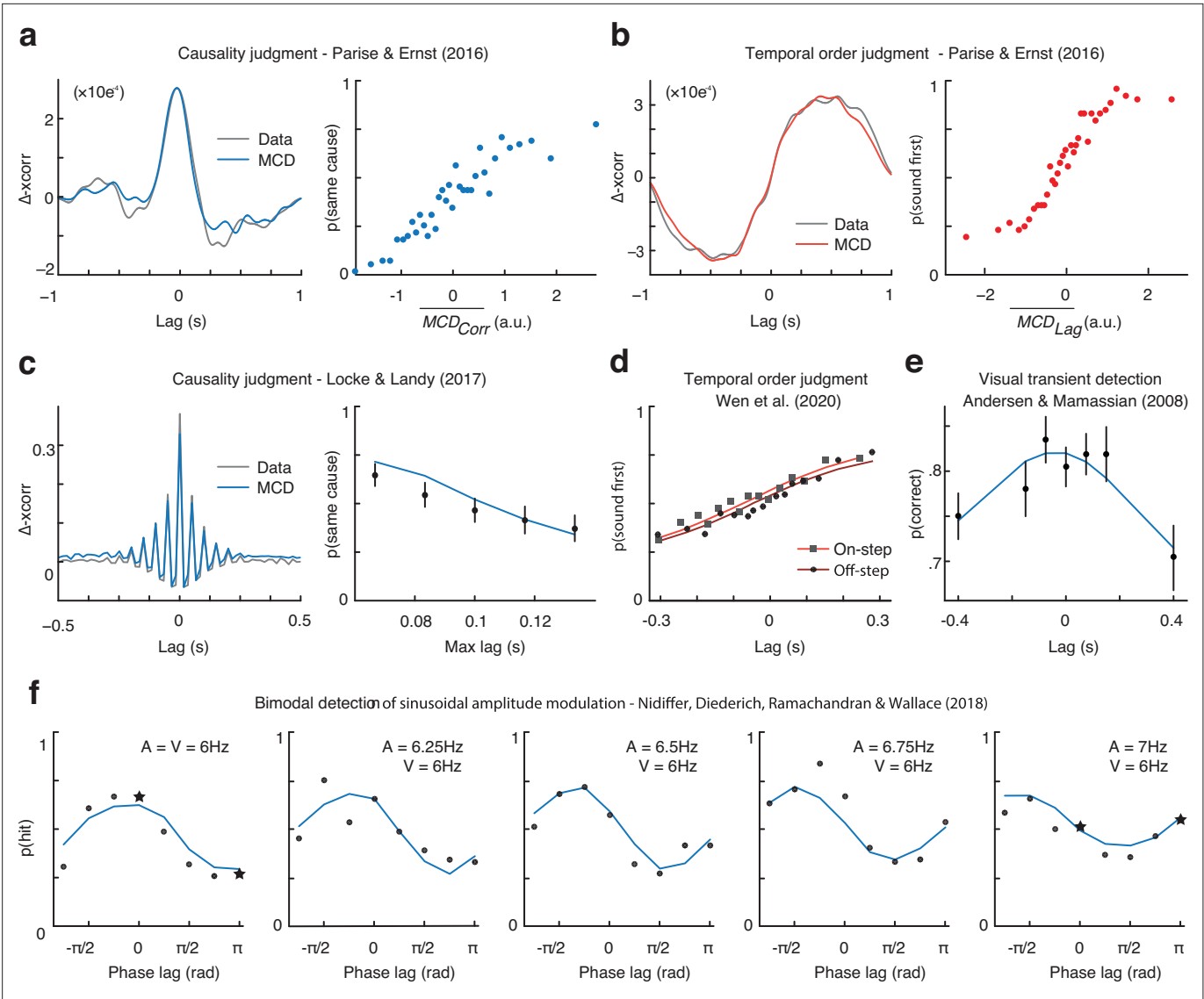

**Figure 6.** Multisensory correlation detector (MCD) simulations of published results. (**a**) Results of the causality judgment task of *Parise and Ernst, 2016*. The left panel represents the empirical classification image (gray) and the one obtained using the MCD model (blue). The right panel represents the output of the model plotted against human responses. Each dot corresponds to 315 responses. (**b**) Results of the temporal order judgment task of *Parise and Ernst, 2016*. The left panel represents the empirical classification image (gray) and the one obtained using the MCD model (red). The right panel represents the output of the model plotted against human responses. Each dot corresponds to 315 responses. (**c**) Results of the causality judgment task of *Locke and Landy, 2017*. The left panel represents the empirical classification image (gray) and the one obtained using the MCD model (blue). The right panel represents the effect of maximum audiovisual lag on perceived causality. Each dot represents on average 876 trials (range = [540, 1103]). (**d**) Results of the temporal order judgment task of *Wen et al., 2020*. Squares represent the onset condition, whereas circles represent the offset condition. Each dot represents ≈745 trials. (**e**) Results of the detection task of *Andersen and Mamassian, 2008*, showing auditory facilitation of visual detection task. Each dot corresponds to 336 responses. (**f**) Results of the audiovisual amplitude modulation detection task of *Nidiffer et al., 2018*, where the audiovisual correlation was manipulated by varying the frequency and phase of the modulation signals. Each dot represents ≈140 trials. The datapoint represented by a star corresponds to the stimuli displayed in *Figure 6—figure supplement 5*.

The online version of this article includes the following figure supplement(s) for figure 6:

**Figure supplement 1.** Stimuli and reverse correlation analyses of *Parise and Ernst, 2016*.

**Figure supplement 2.** Stimuli of *Locke and Landy, 2017*.

**Figure supplement 3.** Stimuli of *Wen et al., 2020*.

**Figure supplement 4.** Stimuli of *Andersen and Mamassian, 2008*.

**Figure supplement 5.** Stimuli of *Nidiffer et al., 2018*.

*Figure 6 continued*

**Figure supplement 6.** Effect of unimodal temporal constants on the goodness of fit (Pearson correlation) between the multisensory correlation detector (MCD) model and the data from our experiments 1 and 2.

model accurately predicts this finding (Pearson correlation = 0.98) without any free parameters. Taken together, the present simulations demonstrate that the MCD model can account for the perception of a common cause between stochastic audiovisual sequences, even when the stimuli have a high temporal rate, and highlight the importance of both similarity in the temporal structure and cross-modal lag in audiovisual integration.

## Temporal order judgment for onset and offset stimuli

To investigate the perceived timing of auditory and visual on- and offsets, *Wen et al., 2020* presented continuous audiovisual noise stimuli with step on- and offsets (*Figure 6—figure supplement 3*). The authors parametrically manipulated the audiovisual lag between either of the onset or the offset of the audiovisual stimulus and asked participants to report the perceived temporal order of the corresponding on- or offset, respectively. Despite large variability across participants, the point of subjective simultaneity systematically differed across conditions, with acoustic stimuli more likely appearing to change before the visual stimuli in the onset compared to the offset condition.

To test whether the model could replicate this finding, we generated audiovisual signals with the same temporal manipulations of audiovisual lag and fed them to the MCD. To minimize the number of free parameters, we ran the simulation using the temporal constants of the filters fitted using the data from experiments 1 and 2 (see above). Moreover, to test whether the MCD alone could predict the PSS shift between the onset and offset condition, we combined the data from all participants and used a single GLM to link $\overline{MCD_{lag}}$ (*Equation 9*) to the TOJ data, irrespective of condition. Therefore, this simulation consisted of just two free parameters (i.e., the slope and intercept of the GLM). Overall, the responses of the MCD were in excellent agreement with the empirical data (Pearson correlation = 0.97) and successfully captured the difference across onset and offset condition (*Figure 6d*). For comparison, the Pearson correlation between the data and the psychometric functions (modeled as cumulative Gaussians, fitted independently for onset and offset conditions) was also 0.97, but required twice as many free parameters (two per condition), and it does not account for the underlying sensory information processing.

## Acoustic facilitation of visual transients' detection

To study the temporal dynamic of audiovisual integration of unimodal transients, *Andersen and Mamassian, 2008*, experiment (2) asked participants to detect a visual transient (i.e., a luminance increment) presented slightly before or after a task-irrelevant acoustic transient (i.e., a sound intensity increment, *Figure 6—figure supplement 4*). The task consisted of a two-interval forced-choice, with the visual stimuli set at 75% detection threshold, and the asynchrony between visual and acoustic transients parametrically manipulated using the method of constant stimuli.

To simulate this experiment, we fed the stimuli to the MCD model and used a GLM to link the $\overline{MCD_{corr}}$ (*Equation 8*) responses to the proportion of correct responses. The temporal constants of the model were constrained based on the results of experiments 1 and 2, so that this simulation had two free parameters: the slope and the intercept of the GLM. Overall, the MCD could reproduce the data of *Andersen and Mamassian, 2008*, and the Pearson correlation between data and model responses was 0.91 (*Figure 6e*).

## Detection of sinusoidal amplitude modulation

To test whether audiovisual amplitude modulation detection depends on the correlation between the senses, *Nidiffer et al., 2018* (experiment 2) asked the participants to detect audiovisual amplitude modulation. Stimuli consisted of a pedestal intensity to which (in some trials) the authors added a near-threshold sinusoidal amplitude modulation (*Figure 6—figure supplement 5a*). To manipulate audiovisual correlation, the authors varied the frequency and phase of the sinusoidal modulation signal. Specifically, the frequency of auditory amplitude modulation varied between 6 Hz and 7 Hz

(five steps), while the phase shift varied between 0 and 360° (eight steps). The frequency and phase shift of visual amplitude modulation were instead constant and set to 6 Hz and 0°, respectively.

Both phase and frequency systematically affected participants' sensitivity; however, the results do not show any evidence of a frequency-doubling effect. This may appear as a surprising finding: given the analogy between this study and our experiment 2, a frequency-doubling effect should be intuitively expected (if, as we claim, correlation detection relied on transient input channels; though see *Figure 6—figure supplement 5a*, for a visualization of the difference between these studies). In theory, when the amplitude modulations in the two modalities are 180° out of phase, the modulation signals are negatively correlated, and negatively correlated signals become positively correlated once fed to unsigned transient channels (*Figure 6—figure supplement 5a*, bottom-left stimuli). When calculating the Pearson correlation of pairs of audiovisual signals, however, we should consider the whole stimuli: including the pedestal, not just the amplitude modulations. Indeed, the stimuli used by *Nidiffer et al., 2018* also consisted of linear ramps at onset and offset and a pedestal intensity level, compared with which the depth of the amplitude modulation was barely noticeable (i.e., the modulation depth was about 6% of the pedestal level, see *Figure 6—figure supplement 5*). Critically, Nidiffer and colleagues did not account for pedestals or onset and offset ramps when computing audiovisual correlation in their original paper. Once the audiovisual correlation is computed while also considering the ramps (and pedestals, see scatterplots in *Figure 6—figure supplement 5a*), the lack of a frequency-doubling effect becomes apparent: all stimuli used by *Nidiffer et al., 2018* are strongly positively correlated, though such a correlation slightly varied across conditions (being 1 when amplitude modulation had the same frequency in both modalities and zero phase shift, and 0.8 in the least correlated condition, see *Figure 6—figure supplement 5a and b*).

To simulate the experiment of *Nidiffer et al., 2018*, we fed the stimuli (including pedestals and onset and offset ramps) to the model and used the $\overline{MCD_{corr}}$ (*Equation 8*) output and a GLM to obtain the hit rate for the detection task. Given that neither the temporal constants of the MCD optimized for the experiments 1 and 2, nor those optimized for *Parise and Ernst, 2016*, provided a good fit for Nidiffer's data, we set both the temporal constants of the MCD and the slope and intercept of the GLM as free parameters. Hence, the model could replicate the results of *Nidiffer et al., 2018*, and the Pearson correlation between the model and human responses was 0.89.

## Discussion

Taken together, the present results demonstrate the dominance of transient over sustained channels in audiovisual integration. Based on the assumption that perceived synchrony across the senses depends on the (Pearson) correlation of the unimodal signals (*Wei, 2006*), we generated specific hypotheses for perceived audiovisual synchrony—depending on whether such a correlation is computed over transient or sustained inputs. Such hypotheses were then tested against the results of two novel psychophysical experiments, jointly showing that Pearson correlation between transient input signals systematically determines the perceived timing of audiovisual events. Based on that, we revised a general model for audiovisual integration, the MCD, to selectively receive input from unimodal transient instead of sustained channels. Inspired by the motion detectors originally proposed for insect vision, such a biologically plausible model integrates audiovisual signals through correlation detection and could successfully account for the results of our psychophysical experiments along with a variety of recent findings in multisensory research. Specifically, once fed with transient input channels, the model could replicate human TOJs, SJs, causality judgments and crossmodal signal detection under a broad manipulation of input signals (i.e., step stimuli, trains of impulses, sinusoidal envelopes, etc.).

Previous research has already proposed a dominance of transient over sustained channels in audiovisual perception, with evidence coming from both psychophysical detection tasks and neuroimaging studies (*Andersen and Mamassian, 2008*; *Werner and Noppeney, 2011*). However, the role of transient and sustained channels on the perceived timing of audiovisual events (arguably the primary determinant of multisensory integration) has never been previously addressed, let alone computationally framed within a general model of multisensory integration. This study fills this obvious gap and explains all such previous findings in terms of the response dynamics of the MCD model, thereby comprehensively demonstrating that audiovisual integration relies on correlated input from unimodal transient channels. Recent research, however, criticized the MCD model for its alleged inability to either process audiovisual stimuli with a high temporal rate or detect signal correlation for short

temporal intervals (*Locke and Landy, 2017*; *Colonius and Diederich, 2020*). By receiving inputs from transient unimodal channels, this revised version of the model fully addresses such criticisms. Indeed, the new MCD can near perfectly predict human performance in a causality judgment task with stimuli with a high temporal rate (up to 18 Hz) using the same set of parameters optimized for stimuli with a much lower rate (5 Hz), with a Pearson correlation coefficient of 0.99. Interestingly, recent results *Parise, 2024* demonstrate that a population of spatially tuned MCD units can also account for the integration of ecological audiovisual stimuli over time and space, thereby replicating phenomena such as the McGurk Illusion, the Ventriloquist Illusion and even attentional orienting.

While MCD simulations could replicate the shape of the empirical psychometric curves, standard psychometric functions can also fit the same datasets, sometimes with even higher goodness of fit. Hence, one might wonder what the advantage of the current modeling approach is. When comparing MCD simulations with psychometric fits, it is important to focus on the differences between these two modeling approaches. Psychometric functions usually relate some physical parameter (the independent variable) to a measure of performance (the dependent variable) based on statistical considerations regarding the stimuli and the underlying perceptual decision-making process. For example, based on assumptions on the nature of the bell-shaped distribution of audiovisual SJs, Yarrow and colleagues (*Yarrow et al., 2011*) proposed a model for SJs that fits the SJ data of our experiment 1 just as well the MCD model (though with a larger number of free parameters). Psychometric curves for SJs, however, are not always bell-shaped: for example, in experiment 2, they are sinusoidal. This finding is naturally captured by the MCD model but not by models that enforce some specific shape for the psychometric functions. The reason is that statistical approaches to psychometric curve fitting are usually blind to the stimuli and agnostic as to how the raw input signals are transformed into evidence for perceptual decision-making. Indeed, while the inputs for psychometric fits are some parameters of the stimuli (e.g., the amount of audiovisual lag), the inputs for the MCD simulations are the actual stimuli themselves. That is, the MCD is a stimulus computable model that extracts from the raw signal the evidence that is then fed into the perceptual decision-making process (i.e., the observer model, *Figure 5a*). By making explicit all the processing steps that link the input stimuli to a button press, the MCD model can tailor its predictions to any input stimuli, with the shape of the psychometric curves being unconstrained in principle, yet fully predictable based only on the input signals and the experimental task. This is why the same MCD model can predict bell-shaped psychometric functions in the SJ task of experiment 1, sigmoidal functions in the TOJ task of experiment 1, and sinusoidal functions in experiment 2, whereas three different functions were necessary for the psychometric fits of the same experiments. Moreover, unlike psychometric fits, the MCD model is a biologically inspired neural model; as such, it not only allows one to account for behavioral responses, but also for neurophysiological data, as recently shown through magnetoencephalography (*Pesnot et al., 2022*).

Previous research attributed the multisensory benefits on perceptual decision-making tasks to 'late', post-sensory changes occurring at the level of the decision dynamics, with evidence stemming from EEG activity in brain regions commonly considered to be involved in 'high-level', decision-making (*Franzen et al., 2020*). While the present study cannot directly address such neurophysiological considerations, the computational framework proposed here, however, challenges any post-sensory interpretations. Indeed, the model proposed in this study can be divided into two separate components: the MCD, which handles the 'early' sensory processing stage, and a 'late' observer model, which receives input from the MCD and uses this information (along with the task demands) to generate a perceptual classification response (i.e., a button-press). In this context, it is important to note how multisensory benefits in detection tasks critically depend on the correlation and timing of audiovisual events: namely, the two factors that mostly affect the responses of the MCD model (e.g., see *Figure 6e and f*). Therefore, it is not surprising to see that the current framework can account for the effects of stimulus timing on performance purely based on the dynamics of the MCD responses, with no need for ad hoc adjustments in the perceptual decision-making process. Further studies will hopefully reconcile such discrepant interpretations on the origins of multisensory benefits on perceptual decision-making tasks, possibly through a better understanding of the computations underlying the electrophysiological correlates recorded with modern imaging techniques.

Although the present study unequivocally supports a dominance of transient channels in multisensory integration, it is still necessary to consider which role, if any, sustained channels may play in

crossmodal perception. Indeed, earlier studies have shown that sustained information, such as the intensity of visual and acoustic stimuli, does indeed systematically affect performance in behavioral tasks (*Stein et al., 1996*; *Odgaard et al., 2004*), and can even elicit phenomena known as crossmodal correspondences (*Parise and Spence, 2013*). The mapping of intensity between vision and audition, however, is a somewhat peculiar one as it is not obvious whether increasing intensity in one modality is mapped to higher or lower intensity in the other (i.e., an obvious mapping between acoustic and lightness intensity, rooted in natural scene statistics, has not been reported, yet). A perhaps more profound mapping between sustained audiovisual information relates to redundant cues, that is, properties or a physical stimulus that can be jointly estimated via two or more senses; such as the size of an object, which can be simultaneously estimated through vision and touch (*Ernst and Banks, 2002*; *van Dam et al., 2014*). This is a particularly prominent aspect of multisensory perception, and the estimated size of an object is arguably sustained, rather than transient stimulus information. While the present study cannot directly address such a question, we propose that the correlation between visual and tactile transients, like those occurring when we reach and make contact with an object, is what the brain needs to solve the correspondence problem, and that hence let us infer that what we see and touch are indeed coming from the same distal stimulus. Then, once made sure that the spatial cues that we get from vision and touch are redundant, size information can be optimally integrated. Testing such a hypothesis is surely a fertile subject for future research.

Finally, it is important to consider what the advantages of transient over sustained stimulus information for multisensory perception are. An obvious one is parsimony as dropping information that does not change over time entails lower transmission bandwidth by minimizing redundancy through efficient input coding (*Barlow, 1961*). Therefore, by only representing input variations, transient channels operate as event detectors, signaling the system of potentially relevant changes in the surrounding. Interestingly, a similar approach has grown in popularity in novel technological applications, such as neuromorphic circuits and event cameras. Indeed, while traditional camera sensors operate on a frame-based approach, whereby inputs are periodically sampled based on an internal clock, event cameras only respond to external changes in brightness as they occur, thereby reducing transmission bandwidth and maximizing dynamic range. Interestingly, recent work has even successfully exploited Hassenstein–Reichardt detectors as a biologically inspired solution for detecting motion with event cameras (*D'Angelo et al., 2020*). Hence, considering the mathematical equivalence of the MCD and the Hassenstein–Reichardt detectors, the present study suggests the intriguing possibility of using the revised MCD model as a biologically inspired solution for sensory fusion in future multimodal neuromorphic systems.

## Materials and methods
### The MCD model
The modified MCD model closely resembles the original model, but it takes input from unimodal transient (instead of sustained) input channels. That is, rather than being simply low-pass filtered, time-varying visual and auditory signals ($S_V(t)$, $S_A(t)$) are independently filtered by band-pass filters ($f$). Following *Adelson and Bergen, 1985*, band-pass filters are modeled as biphasic impulse response functions defined as follows:

$$f_{mod}(t) = \frac{t}{\tau_{mod}} \cdot e^{\frac{-t}{\tau_{mod}}} \cdot \left(1 - \frac{t^2}{\tau_{mod}^2} \cdot n!\right) \tag{1}$$

In this equation, $\tau_{mod}$ is the modality-dependent temporal constant of the filter (mod = [a,v]). Based on the previous results of experiments 1 and 2, we set these constants to be $\tau_V = 0.070$ s and $\tau_A = 0.055$ s for the visual and auditory filters, respectively. In line with *Adelson and Bergen, 1985*, the parameter $n$ (which controls the negative lobe of the impulse response) is set to 3.

As in the original implementation of the MCD model, the low-pass temporal filter of the correlation unit was

$$f_{av}(t) = \frac{t}{\tau_{av}} \cdot e^{\frac{-t}{\tau_{av}}} \tag{2}$$

The temporal constant $\tau_{av}$ was set to 0.674 s. Although the bimodal temporal constant $\tau_{av}$ has a broad temporal tuning, the unimodal temporal constants $\tau_V$ and $\tau_A$ have systematic effects on the goodness of fit of the MCD model to our new psychophysical data. Therefore, in *Figure 6—figure supplement 6* we show how the correlation between the MCD model and empirical data varies as a function of the temporal tuning of the visual and auditory transient detectors.

The filtered unisensory stimuli $Sf_{mod}(t)$ feeding into the correlation unit were obtained as follows:

$$Sf_{mod}(t) = [S_{mod}(t) * f_{mod}(t)]^2 \tag{3}$$

where (*) represents the convolution operator. Filtered signals are squared before being summed to render the responses insensitive to the polarity of changes in intensity (*Stigliani et al., 2017*).

Like in the original MCD model, each subunit ($u_1$, $u_2$) of the detector independently combines filtered visual and auditory signals as follows:

$$u_1(t) = Sf_v(t) . [Sf_a(t) * f_{av}(t)] \tag{4}$$
$$u_2(t) = Sf_a(t) . [Sf_v(t) * f_{av}(t)] \tag{5}$$

To this end, the signals are convolved (*) with the low-pass temporal filters. The response of the subunits is eventually multiplied or subtracted.

$$MCD_{corr}(t) = \sqrt{u_1(t) \cdot u_2(t)} \tag{6}$$
$$MCD_{lag}(t) = u_2(t) - u_1(t) \tag{7}$$

The resulting time-varying responses represent the local temporal correlation ($MCD_{Corr}$) and lag ($MCD_{Lag}$) across the signals. To reduce such time-varying responses into a single summary variable representing the total amount of evidence from each trial, we simply averaged the output of the detectors over a given temporal window of N samples:

$$\overline{MCD_{corr}} = \frac{1}{N} \sum_{t=1}^{N} MCD_{corr}(t) \tag{8}$$

$$\overline{MCD_{lag}} = \frac{1}{N} \sum_{t=1}^{N} MCD_{lag}(t) \tag{9}$$

In the present simulations, the width of the temporal window varies across experiments due to the variable duration of the audiovisual stimuli. The output of the MCD model is eventually transformed into probabilities using a general linear model with a probit link function (assuming additive Gaussian noise; see *Parise and Ernst, 2016*, for a similar approach). A Matlab implementation of the model is available as *Source code 1*.

## Modeling sustained input channels

In line with previous work (*Burr et al., 2009*; *Stigliani et al., 2017*; *Parise and Ernst, 2016*), here we model sustained input channels as low-pass temporal filters with the following impulse response function:

$$f_{mod}(t) = \frac{t}{\tau_{mod}} \cdot e^{\frac{-t}{\tau_{mod}}} \tag{10}$$

where $\tau_{mod}$ is the modality-dependent temporal constant of the filter (mod = [a,v]). Such a low-pass filter has the same shape as the bimodal filters of the MCD (*Equation 2*) and of the unimodal filter of the original MCD model (*Parise and Ernst, 2016*).

## Experiment 1: The relationship between the PSS and WSS measured using TOJs vs. SJs

TOJs and SJs are the two main psychophysical tasks to measure sensitivity to lags across the senses. With both tasks it is possible to estimate the PSS and WSS; however, when measured on the same subjects, the PSS and WSS measured from the two tasks are often not correlated (*García-Pérez and*

*Alcalá-Quintana, 2012*; *Linares and Holcombe, 2014*; *Machulla et al., 2016*). This finding has been sometimes considered evidence for independent underlying neural mechanisms. Given that in experiment 1 we estimated the PSS and WSS with both TOJs and SJs, we can repeat the same analyses on our dataset. *Figure 2—figure supplement 1* shows the scatterplot of the PSS measured with the TOJ against the PSS measured with the SJ, and Figure S1D the scatterplot of the WSS measured with the TOJ against the SJ. Each point corresponds to one psychometric function in *Figure 2—figure supplement 1a*. As in previous studies, the PSS and WSS measured with the two tasks are not significantly correlated (PSS: $r = -0.16$, p=0.38; WSS: $r = 0.16$, p=0.39).

While such a finding intuitively suggests the existence of independent mechanisms underlying the two tasks, our model clearly suggests otherwise. Indeed, the MCD model provides two outputs: one representing the decision variable for the SJ (*Equation 8*) and the other for the TOJ (*Equation 9*). Therefore, we propose that TOJs and SJs share a common mechanism for sensory processing: the MCD model (*Equations 1–7*). However, the following decision-making processes (see *Figure 5a*) are independent across the two tasks, hence the lack of correlation between the PSS and WSS estimated using SJs vs. TOJs.

## Sample size

To properly substantiate our claims and quantitatively test our model, this study relies on a large collection of three novel psychophysical datasets and six previously published ones. These consisted of behavioral responses from a variety of tasks such as TOJs, SJs, causality judgments and detection tasks for a total of 68,693 trials. Our experiments 1 and 2 alone consisted of 12,600 trials (4800 for the TOJ task in experiment 1, 4800 for the SJ task in experiment 1, and 3000 for experiment 2). Given the nature of the present study, we were especially interested in determining the shapes of the psychometric functions, hence we prioritized collecting a large number of trials per observer (over a large pool of observers). Considering that we expected both large effect sizes (see *Figure 1b and c*) and low individual variability, the sample size of our new experiments is more than sufficient to draw reliable conclusions. Single-subject analyses (*Figure 2—figure supplement 1*, *Figure 3—figure supplement 1*) show consistent behavior across participants support our original assumptions.

Nevertheless, it is important to stress that our psychophysical experiments only represent a small fraction of the overall dataset used in this study to assess and model the contribution of transient and sustained channels in multisensory integration. Indeed, when calculating the sample size of this study, we must also include all the previously published datasets that were reanalyzed and simulated with the MCD model. Hence, our conclusions are supported by a large-scale analysis and computational modeling of a vast set of behavioral data, consisting of 68,693 trials from a sample of 110 observers, collectively providing strong converging evidence for the dominance of transient over sustained input channels in multisensory integration (see *Appendix 1—table 1*).

The data from experiments 1 and 2 are available as *Figure 2—source data 1*, *Figure 2—source data 2*, and *Figure 3—source data 1*.

## The quadrature MCD model

The MCD units used so far receive input from unimodal transient channels modeled as a single biphasic temporal filter followed by squaring nonlinearities. In the original version of *Adelson and Bergen, 1985*, however, such a transient detection unit consisted of two biphasic temporal filters applied in parallel to the input, and the resulting signals are then squared and summed to each other. Although for the present simulations such a simplified version of the transient detector was sufficient, this may not be the case for different and more complex sets of stimuli. Hence, for completeness, here we also describe the full transient detector model.

Just like the simplified transient detector (*Equations 1 and 3*), also the full transient detector consists of biphasic temporal filters. However, instead of passing the signal through a single biphasic filter, the full transient detector consists of two biphasic temporal filters 90° out of phase, applied in parallel to the incoming signals. Following Adelson and Bergen, these quadrature filters are modeled as follows:

$$f_n\left(t\right) = \left(\frac{t}{\tau_{bp}}\right)^n \cdot e^{\frac{-t}{\tau_{bp}}} \cdot \left[\frac{1}{n!} - \frac{1}{(n+2)!} \cdot \left(\frac{t}{\tau_{bp}}\right)^2\right] \tag{11}$$

The phase of the filter is determined by $n$, which, based on **Emerson et al., 1992** takes the values of 6 for the fast filter and 9 for the slow one. The temporal constant of the filters is determined by the parameter $\tau_{bp}$.

Fast and slow filters are applied to each unimodal input signal and the two resulting signals are squared and then summed. Then, a compressive nonlinearity (square root) is applied to the output to constrain it within a reasonable range. Therefore, the output of each unimodal unit feeding into the correlation detector takes the following form:

$$Sf_{mod}(t) = \sqrt{\left[S_{mod}(t) * f_6(t)\right]^2 + \left[S_{mod}(t) * f_9(t)\right]^2} \tag{12}$$

where $mod = vid, aud$ represents the sensory modality and $*$ is the convolution operator. These filtered signals are then multiplied in the two subunits of the MCD following the same logic as the simplified model (**Equations 4–9**).

**Figure 6—figure supplement 6** displays how the temporal tuning of the unimodal temporal filter in the quadrature model affects the goodness of fit of the simulation of our experiments 1 and 2. Clearly, the temporal tuning of the unimodal filters strongly affects the goodness of fit of the model: although overall a tuning below 0.06 s is required for a good fit, the resulting landscape is highly irregular and overall favors slightly faster temporal constants for audition than vision. The parameter defining the low-pass temporal filter of the MCD subunits, instead, is less sensitive to the exact tuning. Hence, for this figure its value is arbitrarily set to 0.674 s. The psychometric functions generated by the full quadrature model are very similar to the ones generated by the simplified model (**Figure 5**). A MATLAB implementation is available as **Source code 1**.

## Acknowledgements

We extend gratitude to Prof. Marty Banks, whose early discussions with us helped shape the foundation of this study, and Dr. Irene Senna for her insightful comments throughout. This work was funded by the Deutsche Forschungsgemeinschaft (DFG, German Research Foundation), Project ID 251654672-TRR 161 (Project C05).

## Additional information

### Funding

| Funder | Grant reference number | Author |
| --- | --- | --- |
| Deutsche Forschungsgemeinschaft | 251654672-TRR 161 | Marc O Ernst |

The funders had no role in study design, data collection and interpretation, or the decision to submit the work for publication.

### Author contributions

Cesare V Parise, Conceptualization, Data curation, Software, Formal analysis, Investigation, Visualization, Methodology, Writing – original draft, Writing – review and editing; Marc O Ernst, Conceptualization, Resources, Methodology, Writing – review and editing

### Author ORCIDs

Cesare V Parise (iD) https://orcid.org/0009-0000-6092-561X

### Ethics

The experiment was conducted in accordance to the Declaration of Helsinki and was approved by the ethics committee of the University of Bielefeld. Participants received 8 euros per hour and provided written informed consent before participating to the experiment.

Reviewer #1 (Public Review): https://doi.org/10.7554/eLife.90841.3.sa1
Reviewer #2 (Public Review): https://doi.org/10.7554/eLife.90841.3.sa2

Author response https://doi.org/10.7554/eLife.90841.3.sa3

## Additional files

### Supplementary files

MDAR checklist

Source code 1. MATLAB implementations of the MCD model and the quadrature MCD model.

### Data availability

All data is included as source data files for Figures 2 and 3.

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

# Appendix 1

**Appendix 1—table 1.** Sample size of the datasets modeled and analyzed in the present study. Two of the datasets listed here (i.e., experiment 1 and *Parise and Ernst, 2016*) consisted of two tasks, each tested on the same pool of observers. The last row represents the total number of observers and trials, the average number of trials per participant, and the average correlation between multisensory correlation detector (MCD) simulations and human data. Note how the revised MCD tightly replicated human responses in all of the datasets included in this study, despite major differences in stimuli, tasks, and sample sizes of the individual studies (see last column).

| Dataset | Task | Number of observers | Number of trials | Trials for observer | MCD-data correlation |
|---|---|---|---|---|---|
| | Simultaneity judgment | | 4800 | 600 | 0.99 |
| Experiment 1 | Temporal order judgment | 8 | 4800 | 600 | 0.99 |
| Experiment 2 | Simultaneity judgment | 5 | 3000 | 600 | 0.98 |
| | Causality judgment | | 9300 | 1860 | 0.98 |
| *Parise and Ernst, 2016* | Temporal order judgment | 5 | 9300 | 1860 | 0.99 |
| *Locke and Landy, 2017* | Causality judgment | 10 | 7200 | 720 | 0.99 |
| *Wen et al., 2020* | Temporal order judgment | 57 | 22,337 | ≈392 | 0.97 |
| *Andersen and Mamassian, 2008* | Transient detection | 13 | 2352 | ≈181 | 0.91 |
| *Nidiffer et al., 2018* | Modulation detection | 12 | 5604 | 467 | 0.89 |
| | Overall | 110 | 68,693 | ≈624 | 0.97 |

**Appendix 1—table 2.** Results of Friedman test and Bayesian repeated measures ANOVA for experiment 1.

Four separate Friedman tests were used to assess whether the four experimental conditions differed in terms of point of subjective simultaneity (PSS) and window of subjective simultaneity (WSS) in the temporal order judgment (TOJ) and simultaneity judgment (SJ) tasks. The first column represents the variables, the second the $\chi^2$ value and the degrees of freedom (in brackets), and the third the p-value. Given that we ran four tests on the same dataset, statistical significance should be computed by comparing the p-value against a Bonferroni-adjusted alpha level of 0.0125 (i.e., 0.05/4). The last column represents the Bayes factor $BF_{01}$ in favor of the null hypothesis as calculated using Bayesian repeated measures ANOVA using the statistical software JASP (JASP Team 2024; version 0.18.3) with the default settings. Together with the Friedman test, the present analyses provide further converging evidence for the lack of meaningful differences across the two tasks and four conditions of experiment 1.

| | Friedman test | | Bayesian ANOVA |
|---|---|---|---|
| | $\chi^2$(df) | p-Value | $BF_{01}$ |
| PSS (TOJ) | 6.15 (3) | 0.1045 | 2.833 |
| PSS (SJ) | 2.85 (3) | 0.4153 | 2.237 |
| WSS (TOJ) | 2.55 (3) | 0.4663 | 2.570 |
| WSS (SJ) | 5.55 (3) | 0.1357 | 1.452 |

