## [Editor Report · eLife assessment]

This **important** study evaluates a model for multisensory correlation detection, focusing on the detection of correlated transients in visual and auditory stimuli. Overall, the experimental design is sound and the evidence is **compelling**. The synergy between the experimental and theoretical aspects of the article is strong, and the work will be of interest to both neuroscientists and psychologists working in the domain of sensory processing and perception.

---

## [Referee Report · Reviewer #1 (Public Review)]

The authors present a model for multisensory correlation detection that is based on the neurobiologically plausible Hassenstein Reichardt detector (Parise & Ernst, 2016). They demonstrate that this model can account for human behaviour in synchrony or temporal order judgements and related temporal tasks in two new data sets (acquired in this study) and a range of previous data sets. While the current study is limited to the model assessment for relatively simple audiovisual signals, in future communications, the authors demonstrate that the model can also account for audiovisual integration of complex naturalistic signals such as speech and music.

The significance of this work lies in its ability to explain multisensory perception using fundamental neural mechanisms previously identified in insect motion processing.

Strengths:

(1) The model goes beyond descriptive models such as cumulative Gaussians for TOJ and differences in cumulative Gaussians for SJ tasks by providing a mechanism that builds on the neurobiologically plausible Hassenstein-Reichardt detector.

(2) This model can account for results from two new experiments that focus on the detection of correlated transients and frequency doubling. The model also accounts for several behavioural results from experiments including stochastic sequences of A/V events and sine wave modulations (and naturalistic Av signals such as speech and music as shown in future communications).

---

## [Referee Report · Reviewer #2 (Public Review)]

Summary:

This is an interesting and well-written manuscript that seeks to detail performance on two human psychophysical experiments designed to look at the relative contributions of transient and sustained components of a multisensory (i.e., audiovisual) stimulus to their integration. The work is framed within the context of a model previously developed by the authors and now somewhat revised to better incorporate the experimental findings. The major takeaway from the paper is that transient signals carry the vast majority of the information related to the integration of auditory and visual cues, and that the Multisensory Correlation Detector (MCD) model not only captures the results of the current study, but is also highly effective in capturing the results of prior studies focused on temporal and causal judgments.

Strengths:

Overall the experimental design is sound and the analyses well performed. The extension of the MCD model to better capture transients make a great deal of sense in the current context, and it is very nice to see the model applied to a variety of previous studies.

Comments on the revised version:

In the revised manuscript, the authors have done an excellent job of responding to the prior critiques. I have no additional concerns or comments.

---

## [Author Response]

The following is the authors’ response to the original reviews.

**Public Reviews:**

**Reviewer #1 (Public Review):**
The authors present a model for multisensory correlation detection that is based on the neurobiologically plausible Hassenstein Reichardt detector. It modifies their previously reported model (Parise & Ernst, 2016) in two ways: a bandpass (rather than lowpass) filter is initially applied and the filtered signals are then squared. The study shows that this model can account for synchrony judgement, temporal order judgement, etc in two new data sets (acquired in this study) and a range of previous data sets.Strengths:(1) The model goes beyond descriptive models such as cumulative Gaussians for TOJ and differences in cumulative Gaussians for SJ tasks by providing a mechanism that builds on the neurobiologically plausible Hassenstein-Reichardt detector.(2) This modified model can account for results from two new experiments that focus on the detection of correlated transients and frequency doubling. The model also accounts for several behavioural results from experiments including stochastic sequences of A/V events and sine wave modulations.Additional thoughts:(1) The model introduces two changes: bandpass filtering and squaring of the inputs. The authors emphasize that these changes allow the model to focus selectively on transient rather than sustained channels. But shouldn't the two changes be introduced separately? Transients may also be detected for signed signals.

We updated the original model because our new psychophysical evidence demonstrates the fundamental role of unsigned transient for multisensory perception. While the original model received input from sustained unimodal channels (low-pass filters), the new version receives input from unsigned unimodal transient channels. Transient channels are normally modelled through bandpass filters (to remove the DC and high-frequency signal components) and squaring (to remove the sign). While these may appear as two separate changes in the model, they are, in fact, a single one: the substitution of sustained with unsigned transient channels (for a similar approach, see Stigliani et al. 2017, PNAS). Either change alone would not be sufficient to implement a transient channel that accounts for the present results.

That said, we were also concerned with introducing too many changes in the model at once. Indeed, we simply modelled the unimodal transient channels as a single band-pass filter followed by squaring. This is already a stripped-down version of the unsigned transient detectors proposed by Adelson and Bergen in their classic Motion Energy model. The original model consisted of two biphasic temporal filters 90 degrees out of phase (i.e., quadrature filters), whose output is later combined. While a simpler implementation of the transient channels was sufficient in the present study, the full model may be necessary for other classes of stimuli (including speech, Parise, 2024, BiorXiv). Therefore, for completeness, we now include in the Supplementary Information a formal description of the full model, and validate it by simulating our two novel psychophysical studies. See Supplementary Information “The quadrature MCD model” section and Supplementary Figure S8.

(2) Because the model is applied only to rather simple artificial signals, it remains unclear to what extent it can account for AV correlation detection for naturalistic signals. In particular, speech appears to rely on correlation detection of signed signals. Can this modified model account for SJ or TOJ judgments for naturalistic signals?

It can. In a recent series of studies we have demonstrated that a population of spatially-tuned MCD units can account for audiovisual correlation detection for naturalistic stimuli, including speech (e.g. the McGurk Illusion). Once again, unsigned transients were sufficient to replicate a variety of previous findings. We have now extended the discussion to cover this recent research: Parise, C. V. (2024). Spatiotemporal models for multisensory integration. bioRxiv, 2023-12.

Even Nidiffer et al. (2018) which is explicitly modelled by the authors report a significant difference in performance for correlated and anti-correlated signals. This seems to disagree with the results of study 1 reported in the current paper and the model's predictions. How can these contradicting results be explained? If the brain detects correlation on signed and unsigned signals, is a more complex mechanism needed to arbitrate between those two?

We believe the reviewer here refers to our Experiment 2 (where, like Nidiffer at al. (2018) we used periodic stimuli, not Experiment 1, which consists of step stimuli). We were also puzzled by the difference between our Experiment 2 and Nidiffer et al. (2018): we induced frequency doubling, Nidiffer did not. Based on quantitative simulations, we concluded that this difference could be attributed to the fact that while Nidiffer included on each trial an intensity ramp in their periodic audiovisual stimuli, we did not. As a result, when considering the ramp (unlike in Nidiffer’s analyses), all audiovisual signals used by Nidiffer were positively correlated (irrespective of frequency and phase offset), while our signals in Experiment 2 were sometimes correlated and other times not (depending on the phase offset). This important simulation is included in Supplementary Figure S7; we also have now updated the text to better highlight the role of the pedestal in determining the direction of the correlation.

(3) The number of parameters seems quite comparable for the authors' model and descriptive models (e.g. PSF models). This is because time constants require refitting (at least for some experimental data sets) and the correlation values need to be passed through a response mode (i.e. probit function) to account for behavioural data. It remains unclear how the brain adjusts the time constants to different sensory signals.

This is a deep question. For simplicity, here the temporal constants were fitted to the empirical psychometric functions. To avoid overfitting, whenever possible we fitted such parameters over some training datasets, while trying to predict others. However, in some cases, it was necessary to fit the temporal constants to specific datasets. This may suggest that the temporal tuning of those units is not crystalised to some pre-defined values, but is adjusted based on recent perceptual history (e.g., the sequence of trials and stimuli participants are exposed to during the various experiments).

For transparency, here we show how varying the tuning of the temporal constants of the filters affects the goodness of fit of our new psychophysical experiments (Supplementary Figure S8). As it can be readily appreciated, the relative temporal tuning of the unimodal transient detector was critical, though their absolute values could vary over a range of about 15 to over 100ms. The tuning of the low-pass filters of the correlation detector (not shown here) displayed much lower temporal sensitivity over a range between 0.1s to over 1s.

This simulation shows the impact of temporal tuning in our simulations, however, the question remains as to how such a tuning gets selected in the first place. An appealing explanation relies on natural scene statistics: units are temporally tuned to the most common audiovisual stimuli. Although our current empirical evidence does not allow us to quantitatively address this question, in previous simulations (see Parise & Ernst, 2016, Supplementary Figure 8), by analogy with visual motion adaptation, we show how the temporal constants of our model can dynamically adjust and adapt to recent perceptual history. We hope these new and previous simulations address the question about the nature of the temporal tuning of the MCD units.

(4) Fujisaki and Nishida (2005, 2006) proposed mechanisms for AV correlation detection based on the Hassenstein-Reichardt motion detector (though not formalized as a computational model).

This is correct, Fujisaki and Nishida (2005, 2007) also hypothesized that AV synchrony could be detected using a mechanism analogous to motion detection. Interestingly, however, they ruled out such a hypothesis, as their “data do not support the existence of specialized low-level audio-visual synchrony detectors”. Yet, along with our previous work (Parise & Ernst, 2016, where we explicitly modelled the experiments of Fujisaki and Nishida), the present simulations quantitatively demonstrate that a low-level AV synchrony detector is instead sufficient to account for audiovisual synchrony perception and correlation detection. We now credit Fujusaki and Nishida in the modelling section for proposing that AV synchrony can be detected by a cross-correlator.

Finally, we believe the reviewer is referring to the 2005 and 2007 studies of Fujisaki and Nishida (not 2006); here are the full references of the two articles we are referring to:

Fujisaki, W., & Nishida, S. Y. (2005). Temporal frequency characteristics of synchrony–asynchrony discrimination of audio-visual signals. Experimental Brain Research, 166, 455-464.

Fujisaki, W., & Nishida, S. Y. (2007). Feature-based processing of audio-visual synchrony perception revealed by random pulse trains. Vision Research, 47(8), 1075-1093.

**Reviewer #2 (Public Review):**
Summary:This is an interesting and well-written manuscript that seeks to detail the performance of two human psychophysical experiments designed to look at the relative contributions of transient and sustained components of a multisensory (i.e., audiovisual) stimulus to their integration. The work is framed within the context of a model previously developed by the authors and is now somewhat revised to better incorporate the experimental findings. The major takeaway from the paper is that transient signals carry the vast majority of the information related to the integration of auditory and visual cues, and that the Multisensory Correlation Detector (MCD) model not only captures the results of the current study but is also highly effective in capturing the results of prior studies focused on temporal and causal judgments.Strengths:Overall the experimental design is sound and the analyses are well performed. The extension of the MCD model to better capture transients makes a great deal of sense in the current context, and it is very nice to see the model applied to a variety of previous studies.Weaknesses:My one major issue with the paper revolves around its significance. In the context of a temporal task(s), is it in any way surprising that the important information is carried by stimulus transients? Stated a bit differently, isn't all of the important information needed to solve the task embedded in the temporal dimension? I think the authors need to better address this issue to punch up the significance of their work.

In hindsight, it may appear unsurprising that transient signals carry most information for audiovisual integration. Yet, so somewhat unexpectedly, this has never been investigated using perhaps the most diagnostic psychophysical tools for perceived crossmodal timing; namely temporal order and simultaneity judgments–along with carefully designed experiments with quantitative predictions for the effect of either channel. The fact that the results conform to intuitive expectations further supports the value of the present work: grounding empirically with what is intuitively expected. This offers solid psychophysical evidence that one can build on for future advancements. Importantly, developing a model that builds on our new results and uses the same parameters to predict a variety of classic experiments in the field, further supports the current approach.

If “significance” is intended as shaking previous intuitions or theories, then no: this is not a significant contribution. If instead, by significance we intend to build a solid empirical and theoretical ground for future work, then we believe this study is not significant, it is foundational. We hope that this work's significance is better captured in our discussion.

On a side note, there is an intriguing factor around transient vs. sustained channels: what matters is the amount of change, not the absolute stimulus intensity. Previous studies, for example, have suggested a positive cross modal mapping between auditory loudness and visual lightness or brightness [Odegaard et al., 2004]. This study, conversely, challenges this view and demonstrates that what matters for multisensory integration in time is not the intensity of a stimulus, but changes thereof.

In a more minor comment, I think there also needs to be a bit more effort into articulating the biological plausibility/potential instantiations of this sustained versus transient dichotomy. As written, the paper suggests that these are different "channels" in sensory systems, when in reality many neurons (and neural circuits) carry both on the same lines.

The reviewer is right, in our original manuscript we glossed over this aspect. We have now expanded the introduction to discuss their anatomical basis. However, we are not assuming any strict dichotomy between transient and sustained channels; rather, our results and simulations demonstrate that transient information is sufficient to account for audiovisual temporal integration.

**Recommendations for the authors:**

**Reviewer #1 (Recommendations For The Authors):**
(1) Related to point 2 of the public review, can the authors provide additional results showing that the model can also account for naturalistic signals and more complex stochastic signals?

While working on this manuscript, we were also working in parallel on a project related to audiovisual integration of naturalistic signals. A pre-print is available online [Parise, 2024, BiorXiv], and the related study is now discussed in the conclusions.

(2) As noted in the public review, Fujisaki and Nishida (2005, 2006) already proposed mechanisms for AV correlation detection based on the Hassenstein-Reichardt motion detector. Their work should be referenced and discussed.

We have now acknowledged the contribution of Fujisaki and Nishida in the modelling section, when we first introduce the link between our model and the Hassenstein-Reichardt detectors.

(3) Experimental parameters: Was the phase shift manipulated in blocks? If yes, what about temporal recalibration?

To minimise the effect of temporal recalibration, the order of trials in our experiments was randomised. Nonetheless, we can directly assess potential short-term recalibration effects by plotting our psychophysical responses against both the current SOA, and that of the previous trials. The resulting (raw) psychometric surfaces below are averaged across observers (and conditions for Experiment 1). In all our experiments, responses are obviously dependent on the current SOA (x-axis). However, the SOA of the previous trials (y-axis) does not seem to meaningfully affect simultaneity and temporal order judgments. The psychometric curves above the heatmaps represent the average psychometric functions (marginalized over the SOA of the previous trial).

All in all, the present analyses demonstrate negligible temporal recalibration across trials, likely induced by a random sequence of lags or phase shifts. Therefore, when estimating the temporal constants of the model, it seems reasonable to ignore the potential effects of temporal recalibration. To avoid increasing the complexity of the present manuscript, we would prefer not to include the present analyses in the revised version.

**Author response image 1. sa3fig1:** Effect of previous trial. Psychometric surfaces for Experiments 1 and 2 plotted against the lag in the current vs. the previous trial. While psychophysical responses are strongly modulated by the lag in the last trial (horizontal axis), they are relatively unaffected by the lag in the previous trial (vertical axis).

(4) The model predicts no differences for experiment 1 and this is what is empirically observed. Can the authors support these null results with Bayes factors?

This is a good suggestion: we have now included a Bayesian repeated measures ANOVA to the analyses of Experiment 1. As expected, these analyses provide further, though mild evidence in support for the null hypothesis (See Table S2). For completeness, the new Bayesian analyses are presented alongside the previous frequentist ones in the revised manuscript.